# Identification of Student Behavioral Patterns in Higher Education Using K-Means Clustering and Support Vector Machine

**Nur Izzati Mohd Talib, Nazatul Aini Abd Majid * and Shahnorbanun Sahran**

Center for Artificial Intelligence Technology, Faculty of Information Science & Technology, Universiti Kebangsaan Malaysia, Bangi 43600, Malaysia
* Correspondence: nazatulaini@ukm.edu.my

**Abstract:** In many academic fields, predicting student academic success using data mining techniques has long been a major research issue. Monitoring students in higher education institutions (HEIs) and having the ability to predict student performance is important to improve academic quality. The objective of the study is to (1) identify features that form clusters that have holistic characteristics and (2) develop and validate a prediction model for each of the clusters to predict student performance holistically. For this study, both classification and clustering methods will be used using Support Vector Machine (SVM) and K-means clustering. Three clusters were identified using K-means clustering. Based on the learning program outcome feature, there are primarily three types of students: low, average, and high performance. The prediction model with the new labels obtained from the clusters also gained higher accuracy when compared to the student dataset with labels using their semester grade.

**Keywords:** student academic performance; identification of student behavioral pattern; higher education; machine learning; support vector machine; k-means clustering

## 1. Introduction

Predicting student performance holistically is important not only to curb the symptoms of students at high risk of failure but also to produce holistic students that are doing well academically and also have the soft skills that will help them after finishing their studies. The ability to predict student performance, especially at an early stage, is important, as lecturers and universities can detect high-risk students earlier. Studies like Stapa et al. [1] have focused on analyzing students and their learning environment to develop a suitable approach for teaching and learning resources. Therefore, preventive measures and providing solutions to students can be given early on, which has been performed by Zainuddin et al. [2]. If this problem can be tracked early, students who are likely to obtain a semester grade point average (SGPA) of less than 2.5; for example, in the Faculty of Technology and Information Science, The National University of Malaysia (UKM) can be given an appropriate rehabilitation program by the responsible party, especially in the field of programming. It often happens that this program is only based on student SGPA. The impact of creating an early detection system has also been performed by Berens et al. [3] and can be helpful for lecturers and universities to have a better understanding of their students. Therefore, a strategy needs to be planned based on Industrial Revolution 4.0 technology, such as machine learning and statistics, to predict the performance of students in Malaysia to produce holistic students who are intellectually brilliant and also possess great characteristics. Universities' primary objective isn't just to ensure student academic success; it's also to improve graduates' employability by utilizing the acquired knowledge [4,5]. Producing holistic graduates is one of the aspirations of national education to guarantee quality national assets.

Academic characteristics are often used in the prediction of student performance, but not many focus on holistic competence characteristics that include leadership, teamwork, critical thinking, communication, problem-solving, integrity, global competence, information literacy, resilience, and others [6]. Since soft skills are crucial for a student's future success, it is important to concentrate on analyzing undergraduate students while they are enrolled in college [7,8]. Students may be enrolled in the same course, however, they have distinct personality traits or learning styles that may contribute to their performance [9–11]. Developing a holistic prediction method is difficult because the holistic characteristics of a student are complex, not static, and involve more than one characteristic. Therefore, it is also important to know the specific characteristics of students that affect student performance. A new method is needed to analyze multiple characteristics, including holistic competencies, and account for changes in those characteristics over time. Based on previous studies, the most common factors that influence a student's performance are demographic, academic, psychological, and institutional [12,13]. According to Francis et al. [12] and Aggarwal et al. [14], using a combination of different factors has been proven to produce different results in terms of prediction accuracy. It was proven by both studies that the combination of academic and non-academic does increase the accuracy of the prediction model significantly. Therefore, finding the optimum features is important in obtaining a more accurate result. In addition to that, analyzing those features across time helps in detecting a trend in students' characteristics related to their performance as shown by Asif et al. [15] and Abdelhafez et al. [16]. Some research has added time components in their prediction by analyzing students either yearly [15], per semester, or weekly [16,17]. For this study, a student will be analyzed weekly throughout their first semester to gain a better understanding of students' characteristics early in their studies.

Preliminary predictions using statistical techniques and data mining algorithms have attracted the attention of researchers because of the potential to predict student performance based on the relationship of variables or characteristics involved in monitoring.

Various data mining methods have been employed in numerous earlier research to predict student performance, and it fits into one of the two main categories, which is supervised learning or unsupervised learning. In research conducted by Francis et al. [12], both methods are combined. The combined methods have outperformed when compared to the other available algorithms, with an accuracy of 75.47%. As a result, in this study, a combination of classification and clustering methods will be used, which are Support Vector Machine (SVM) and K-means clustering for student performance prediction.

The objective of the study is to (1) identify features that form clusters that have holistic characteristics and (2) develop and validate a prediction model for each of the clusters to predict student performance holistically. The research methodology includes the following phases: (1) preparation and collection of target data sets, (2) clustering students into three clusters using K-means clustering, (3) application of data mining classification algorithm, and (4) evaluation of the results. A new model to analyze students holistically is expected to be produced to help the government produce holistic students by reducing the rate of students who fail not only academically but also psychologically.

## 2. Related Works

The predictive model holistically uses academic and non-academic characteristics. Most prediction models have used a combination of different domains; however, not much research has been focused on improving the accuracy of models with specific domains. In the study of Aggarwal et al. [14], model accuracy was compared when using both academic and non-academic data and only academic data. It has been shown that the combination of both academic and non-academic data can obtain better results for prediction models. The highest F1 Score obtained using all parameters is 93.8% (using the Random Forest classifier), which is much higher than the maximum F1 Score obtained using academic parameters only, which is 79.6% (using Logistic Regression, multilayer perceptron, and voting meta classifiers).

The study of Francis et al. [12] found that having more than one domain combination obtains better results than when only one domain is used. Domains that exist in the dataset are demographic, academic, behavioral, and additional characteristics. This research analyzes different domain combinations and the accuracy obtained by the prediction model. This study also shows that one domain by itself, without the combination of other domains, does not obtain high accuracy. The model obtained the highest accuracy when using a combination of academic, behavioral, and additional characteristics. The model obtained an accuracy of 75.47% with that combination.

Referring to Table 1, most studies have evaluated student performance with their scores. Student performance has been evaluated by predicting the final cumulative grade point average (CGPA) [15,18,19]. In addition, some studies predict students categorically according to their scores, i.e., high or low performance [20,21] or high, medium, and low performance [12,15,19,22] pass or fail [23,24]. In the study of Xu et al. [25], student performance was evaluated by the mean score of the course and the range of student positions in the course, which was divided into five categories, namely those who scored the top 20% as the first category and the last 20% as the fifth category. Next, in the study of Gray et al. [23], student performance is defined by the student's academic standing at the end of the year, whether the student passes, fails, fails conditionally, or has to repeat the semester/year. Meanwhile, there are some studies evaluating student performance and whether students complete the course [14] or their studies [3,16].

**Table 1.** Studies related to Prediction of Student Performance.

| References | Attributes | Prediction Model Output |
| --- | --- | --- |
| [15] | Academic | CGPA, Student Category |
| [25] | Academic, Demographic, Psychological | Mean Score for Course, Student Category |
| [3] | Academic, Demographic, Institutional | Complete Studies, Did Not Complete Studies |
| [22] | Demographic, Institutional, Psychological | Student Category |
| [26] | Demographic, Institutional, Psychological | Fail, Pass |
| [27] | Academic, Psychological | Course Marks |
| [28] | Academic, Demographic | CGPA |
| [29] | Academic, Demographic, Institutional, Psychological | Fail, Pass |
| [12] | Demographic, Institutional, Psychological | Student Category |
| [21] | Academic | Low, High Performance |
| [20] | Academic, Demographic, Psychological | Low, High Performance |
| [18] | Academic, Institutional | Final CGPA |
| [24] | Demographic | Fail, Pass |
| [23] | Psychological, Institutional | Academic standing |
| [14] | Academic, Demographic, Institutional | Drop out or complete the course |
| [19] | Academic, Psychological | Final CGPA dan Student Category |
| [8] | Psychological | Student Category |
| [16] | Academic | Finish Studies or Drop-out |

Data mining methods can be divided into classification and clustering. Commonly used algorithms are shown in Table 2. For classification, the most commonly used are naïve bayes (NB), decision tree (DT), neural network (NN), ensemble learning, SVM, and Logistic Regression (LR). Ensemble learning consists of Random Forest, bagging, voting, and boosting. Most studies have tried more than one algorithm for the prediction model and made a comparison of the accuracy of those algorithms. Meanwhile, there have also been studies that prioritize the improvement of one algorithm or the importance of

features or parameters instead of prediction accuracy. Studies like Francis et al. [12] and Asif et al. [15] have combined both classification and clustering methods.

**Table 2.** Types of machine learning algorithm used by current studies.

| References | NB | DT | NN | Ensemble | SVM | LR | Clustering |
|:---:|:---:|:---:|:---:|:---:|:---:|:---:|:---:|
| [15] | / | / | / | / | x | x | / |
| [25] | x | x | / | x | x | x | x |
| [3] | x | / | / | / | x | x | x |
| [22] | x | x | / | x | x | x | x |
| [26] | x | x | x | x | x | x | / |
| [27] | x | x | x | x | x | / | x |
| [28] | x | x | / | x | x | x | x |
| [29] | x | x | x | x | x | / | x |
| [12] | / | / | / | / | / | x | / |
| [21] | / | / | / | / | / | x | x |
| [20] | x | / | / | x | / | x | x |
| [18] | / | / | / | / | x | / | x |
| [24] | x | / | x | x | x | x | x |
| [23] | / | / | / | / | x | x | x |
| [14] | x | / | / | / | / | / | x |
| [19] | / | / | / | / | x | x | x |
| [8] | x | x | x | x | x | x | / |
| [16] | / | / | / | x | / | / | x |

Note: x is No; / is Yes.

In the study of Francis et al. [12], both classification and clustering are combined to create a hybrid algorithm which is K-means clustering with majority voting. Classification is used to determine important characteristics then students are grouped based on those important characteristics using K-means-based clustering. The hybrid model surpassed the accuracy of NB, SVM, DT, and NN models. Clustering is useful for predicting student performance, and some studies plan to improve their work in the future by using the cluster technique because it will help the cold start problem when used in the initial phase to categorize students Mondal et al. [22].

Furthermore, in the study by Asif et al. [15], the clustering in the data can be used to identify the average student development pattern across a number of years. Students were clustered year-by-year using X-means clustering over the four years according to their final marks for each course rather than all years together. This approach also allows atypical students to be identified. Therefore, the time dimension is significant in predicting student performance. Many studies are limited to modeling data obtained at a single point in time. Therefore, a study is needed to examine whether data changes on one factor across semesters, such as learning efficiency characteristics, have a better effect in detecting early students who are at risk of failing that semester.

## 3. Methodology

This section describes the proposed methodology used to classify and cluster student data to predict students' academic performance. The main phases are data preparation, data cleaning, clustering, and classification. As follows, each step of the suggested process is explained below:

### 3.1. Data Preparation

The dataset used in this study was obtained from the Faculty of Technology and Information Science at the National University of Malaysia. The dataset consists of information from 200 students in their first semester of the first year during the 2019/2020 session with records of their characteristics based on their demographic, academic, and behavior with 41 attributes. All features which are used to build the model of prediction are listed in the tables below.

The input features can be divided into four sections which are Part A (Learning Style), Part B (Self-efficacy), Part C (Demographic), and Part D (learning program outcome). Referring to Tables 3–5, information was obtained from the study of Sabri et al. [8]. This paper will improve his research, and therefore, new features of a holistic nature will be added. Table 3 is the research questionnaire for Part A (Learning Style), and Table 4 is the research questionnaire for Part B (Self-efficacy). For Parts A and B, these questions were handed out weekly from week 6 to week 9. Both sections are categorical and require students to pick an option most suitable for them. A five-point scale is used in Part B ranges from Strongly disagree (1), Disagree (2), Neutral (3), Agree (4) to Strongly agree (5). Next, Table 5 includes the demographic information of the students. Lastly, Table 6 includes new features obtained from the UKM system based on the learning program outcome score or known as Hasil Program Pelajar (HPP) as used by Wahab et al. [30] and Othman et al. [5]. The values for this study were obtained from the assignment and lab marks that were then calculated by the system to measure their learning program outcome. Therefore, values for the learning program outcome are between 0.00 to 4.00. The Faculty of Technology and Information Science's learning program outcome was employed in this study. There are eight different types of HPP; however, only HPP1, HPP2, HPP3, HPP4, HPP6, and HPP7 were measured during the student's first semester. At the same time, the output feature is the student's SGPA.

**Table 3.** Survey Questions for Part A (Learning Style).

| Variable | Question | Values |
|---|---|---|
| X1 | What is your level of involvement during last week's lecture? | Very active, Active, Less active, Not active |
| X2 | What is your level of focus during last week's lecture? | Very focus, Focus, Moderate, Less focus, Not focus |
| X3 | How much have your learned from last week's lecture? | Very high, High, Moderate, Low, Very low |
| X4 | What would be your course of action if you had questions during lecture? | Ask the lecturer, Ask a friend, Find your own information, Silent |
| X5 | How would you describe the conditions of last week's lecture room? | Very condusive, Condusive, Less condusive, Not condusive |
| X6 | How much time would you allocate for learning, including doing revision, assignment, exercise, discussion and reading? (Interaction during lecture/lab/tutorial is not considered) | <1 h a week, 1–2 h a week, 3–4 h a week, 5–6 h a week, >6 h a week |
| X7 | When did you complete the exercise/homework/assignment given during last week's lecture? | A day after lecture, 2–3 days after lecture, 4–5 days after lecture, A day before next lecture |
| X8 | When was the most effective learning period for you during the entirety of last week? | In the morning, In the evening, At night, After midnight |
| X9 | Where did you conduct revisions on the subject over the past week? | Residential college, Library, Others |

Note. From Sabri et al. [8].

**Table 4.** Survey Questions for Part B (Self-efficacy).

| Variable | Question |
|---|---|
| SE1 | I ask questions during lecture |
| SE2 | I respond to questions during lecture |
| SE3 | I prepare study schedule and study plan |
| SE4 | I ask and seek help from lecturers |
| SE5 | I make additional lecture notes |
| SE6 | I plan my time for examination/test/quiz |
| SE7 | I ask for help from my friends if I face problems or confusion with regards to any particular topic |
| SE8 | I give my best during examination/test/quiz |
| SE9 | I involve myself in academic discussions with friends |
| SE10 | I take note of/accept assignment feedback |
| SE11 | I can explain subject-related contents to my friends |
| SE12 | I try to answer questions well the first time |
| SE13 | I always meet assignment deadlines |
| SE14 | I try to meet the deadlines of group assignments |
| SE15 | I pay attention during each lecture |
| SE16 | I would make it clear in class if I do not understand any part of the lecture |
| SE17 | I feel nervous whenever I am doing a presentation |
| SE18 | I would volunteer myself to present during any group assignment |
| SE19 | I am confident that I can complete my studies within four years |
| SE20 | I take note of/accept feedback that I receive for any examination/test |

Note. From Sabri et al. [8].

**Table 5.** The information on the demographic features.

| No. | Attributes | Values |
|---|---|---|
| 1 | Program | Computer Science, Software Engineering Multimedia, Software Engineering Information Science, Information Technology |
| 2 | Gender | Female, Male |
| 3 | Race | Malay, Indian, Chinese, and Others |
| 4 | Highest Education | Matriculation, STPM, Asasi, Diploma, International Baccalaureate, Others |
| 5 | Type of Previous Secondary School/High School | Eight different types of schools |
| 6 | Stream of Study | Pure Science, Technical, Art, Religious, Psychical Science, Technical Science, Humanities, Physics, Chemistry, ICT, Computer Science |

### 3.2. Data Pre-Processing

There are six pre-processing methods used, as shown in Figure 1: data collection and integration, data filtering, data cleaning, data transformation, and attribute selection. After the data were collected, the different dataset was then integrated based on the student number, and overall, the dataset has a size of 200. For the data cleaning, if there were too many missing values for certain features, the student was removed from the dataset, as this probably occurred if the student missed providing the information by not answering one of the surveys. This results in a lot of missing values in a certain section. The students

who did not answer one of the surveys were removed entirely. However, if there are only a few missing values, they will be filled in by the mean value. In addition to that, duplicated information was also removed. After cleaning, the dataset is reduced to 140 students.

**Table 6.** The information on the learning program outcome features.

| Learning Program Outcome | Description |
|---|---|
| HPP1 | Fundamental knowledge |
| HPP2 | Computer skills |
| HPP3 | Social responsibility |
| HPP4 | Professionalism and ethics |
| HPP5 | Leadership and teamwork skills |
| HPP6 | Critical thinking and problem-solving |
| HPP7 | Information management skills and lifelong learning principles |
| HPP8 | Management and entrepreneurial skills |

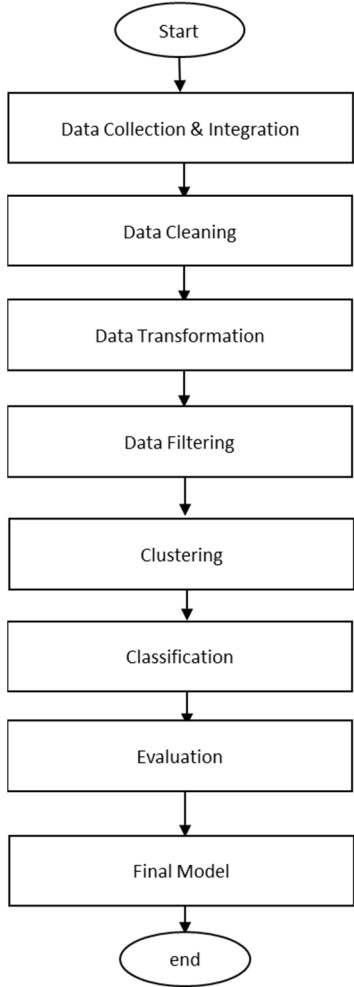

**Figure 1.** Flowchart of the proposed methodology.

Most of the values in the dataset are nominal except for the learning program outcome features. Therefore, for data transformation, the nominal value was converted into numerical values. This is important as machine learning requires numerical values for the prediction model. In addition to that, StandardScaler was used to normalize this data; this is to ensure the individual features resemble normally distributed data. This is important before performing machine learning to ensure it performs well. Before proceeding to

machine learning, the dataset was filtered based on the week. This is to see the difference in students' characteristics each week.

### 3.3. Clustering

K-means clustering is used for this study. The decision of the cluster number is based on the inertia and the "elbow rule". Therefore, using that method, the elbow seems to be between three to five, as shown in Figure 2. For this research, the number of clusters decided is three. After clustering, to determine if the three clusters of data have a significant difference between the mean value, we applied one-way ANOVA for statistical analysis. One-way ANOVA was performed using python's function from the SciPy library.

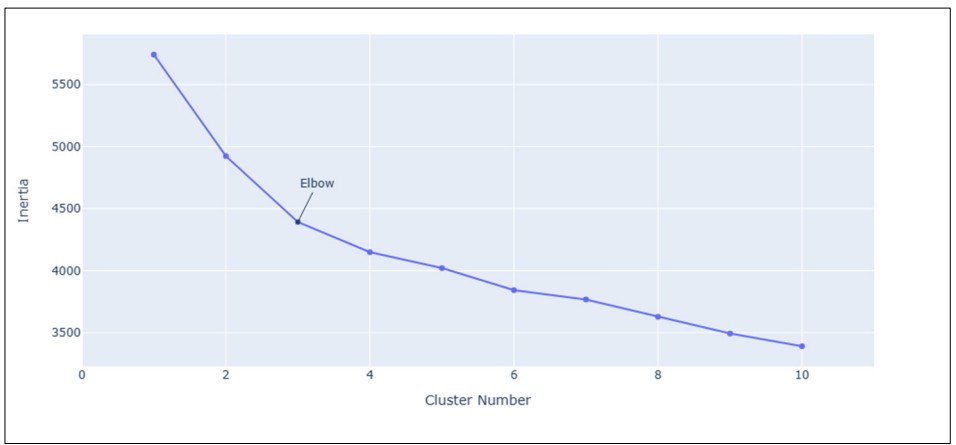

**Figure 2.** K-means inertia versus the number of clusters.

### 3.4. Classification

This study involves binary classification and uses SVM to predict students with high or low performance. SVMs are a group of supervised learning techniques for classifying data, performing regression analysis, and identifying outliers. One advantage of employing SVM is that it works well in high-dimensional environments, even when there are more dimensions than samples [31,32].

For this classification, it will be using a binary class, 0 (low performance) and 1 (high performance), based on the SGPA. If the SGPA is below B+, it is considered low performance, while B+ above is high performance. Then, to make sure the predictive model was producing accurate results, we assessed the performance using a variety of metrics, such as classification accuracy, precision, recall (sensitivity), and f-measure. The following is a list of the equations for accuracy, recall, precision, and F-score:

$$Accuracy = \frac{True\ Positive + True\ Negative}{True\ Positive + False\ Positive + False\ Negative + True\ Negative} \tag{1}$$

$$Recall = \frac{True\ Positive}{True\ Positive + False\ Negative} \tag{2}$$

$$Precision = \frac{True\ Positive}{True\ Positive + False\ Positive} \tag{3}$$

$$F1\ score = 2\frac{Recall * Precision}{Rceall + Precision} \tag{4}$$

## 4. Results

### 4.1. Clustering

The student dataset is clustered into three groups based on each week, as shown in Figures 3 and 4. Figures 3 and 4 show the students by the mean of their characteristics.

Cluster 0 represents the students with a low learning outcome, and Cluster 1 represents the average learning outcome. While Cluster 2 represents those with high learning outcome value. Based on the learning outcome, the self-efficacy and learning style is compared throughout the weeks.

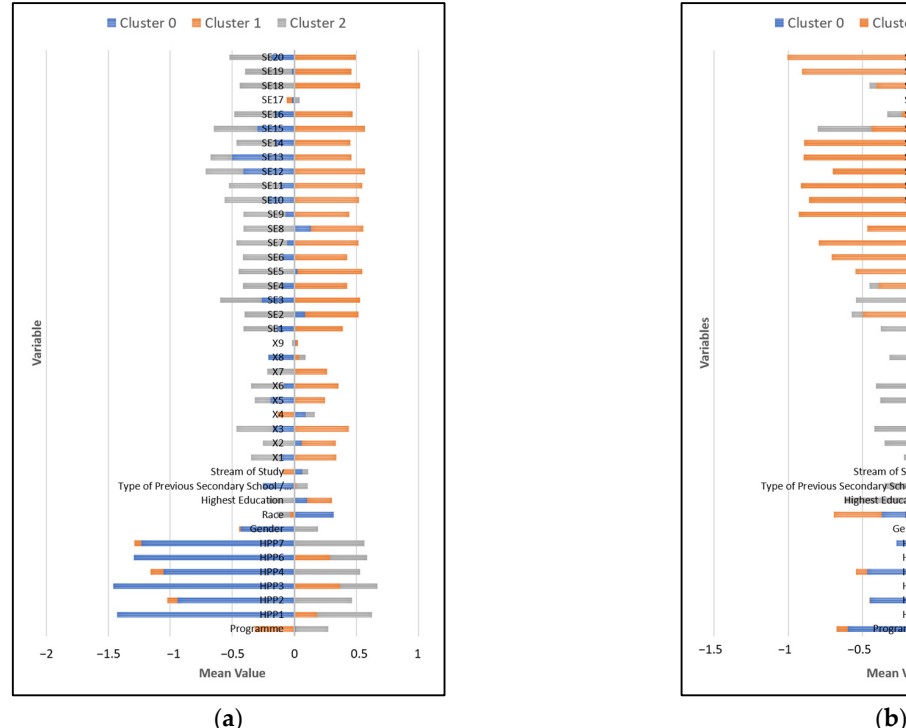

**Figure 3.** Student Cluster using K-means Clustering: (**a**) Week 6; (**b**) Week 7.

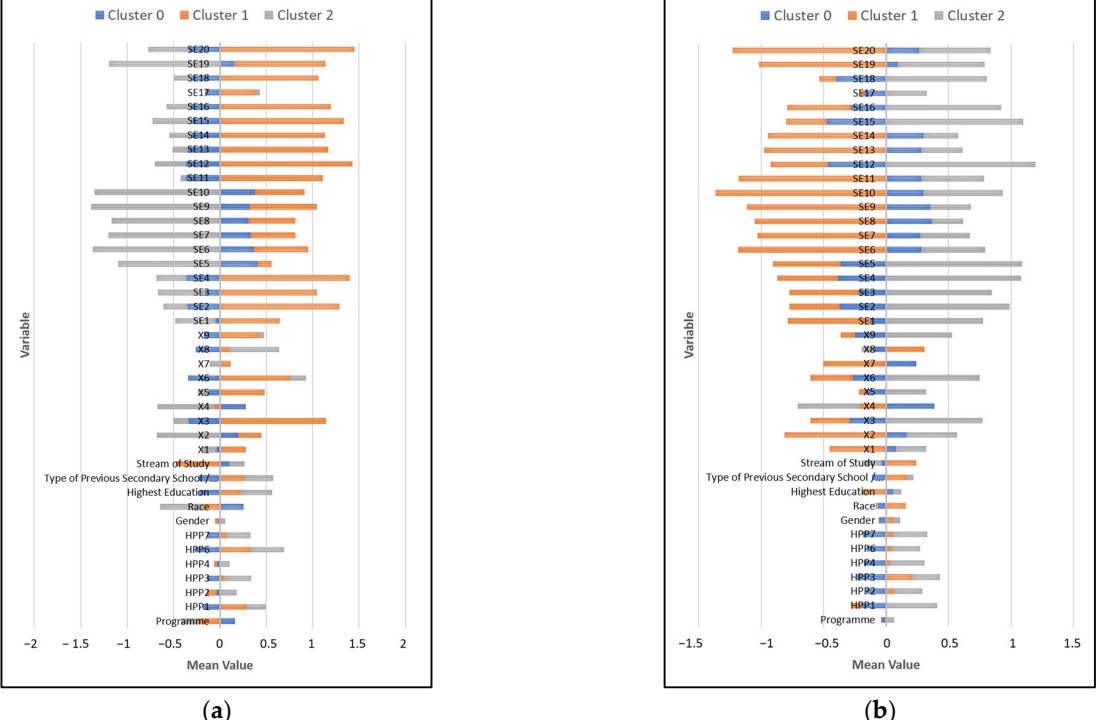

**Figure 4.** Student Cluster using K-means Clustering: (**a**) Week 8; (**b**) Week 9.

The self-efficacy and learning style are compared over the course of the weeks based on the learning outcome in Figure 5. Self-efficacy and learning style has been classified as low (1), average (2), and high (3) using the mean values in Figures 3 and 4. Here, the student's behavioral pattern is identified. Referring to Figure 5, those with high learning program outcomes at the beginning of the semester at week 6 had low self-efficacy and learning style, but by week 9, they had high self-efficacy and learning style. Meanwhile, the student with average learning outcome values, however, fluctuated, beginning with high self-efficacy and learning style at week 6, but had low values for week 7. It increased at week 8 but declined afterward. Lastly, the students with a low learning outcome began with average self-efficacy and learning style values. At week 7, it had a high mean value for self-efficacy and learning style, but by weeks 8 and 9, it declined back to an average mean value.

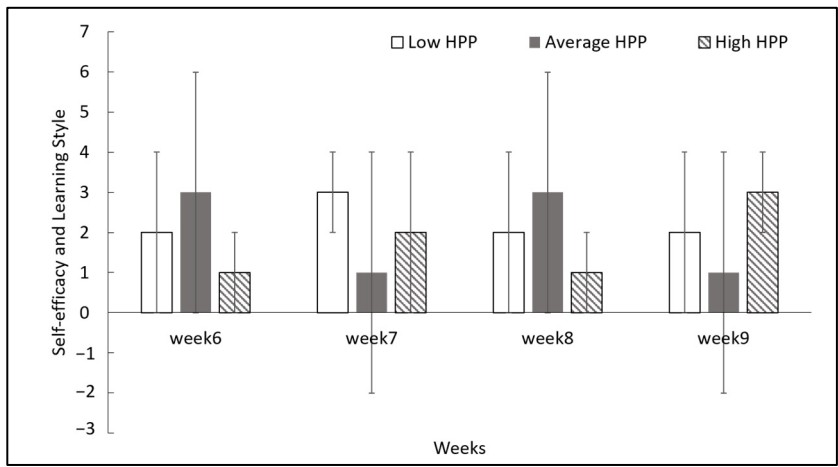

**Figure 5.** Self-efficacy and Learning Style of each week based on mean value.

After dividing the students into three groups, to verify if the three groups of data have a significant difference, we applied one-way ANOVA. As shown in Table 7, The F-value and *p*-value turn out to be equal to 0.79109 and 0.45393, respectively. The *p*-value obtained from ANOVA analysis is greater than 0.05. Therefore, there is no statistically significant difference between the three clusters.

**Table 7.** ANOVA test between clusters.

| Source of Variation | Degree of Freedom | Sum of Squares | Mean Square | F-Value | *p*-Value |
| --- | --- | --- | --- | --- | --- |
| Between Group | 2.0 | 0.365387 | 0.182693 | 0.79109 | 0.45393 |
| Within Group | 489.0 | 112.929007 | 0.230939 | | |
| Total | 491.0 | 113.294394 | | | |

### 4.2. Classification

Before undergoing K-means clustering, the dataset had a label that was low- and high-performance, which was based on the SGPA. There will be a comparison made before and after clustering of the prediction model using the SVM algorithm with the evaluation metrics: accuracy, precision, recall, and f-measure will be used, as shown in Tables 8 and 9. After identifying the three clusters through K-means clustering, the new label obtained from the clustering is used to develop a prediction model. The accuracy before and after clustering is compared in Figure 6.

**Table 8.** Evaluation metrics for the prediction model before clustering.

| Metric | Week | | | |
|---|---|---|---|---|
| | **6** | **7** | **8** | **9** |
| Accuracy | 0.83 | 0.90 | 0.76 | 0.86 |
| Precision | 0.81 | 0.91 | 0.78 | 0.85 |
| Recall | 0.83 | 0.90 | 0.76 | 0.86 |
| F-Measure | 0.80 | 0.89 | 0.77 | 0.85 |

**Table 9.** Evaluation metrics for the prediction model after clustering.

| Metric | Week | | | |
|---|---|---|---|---|
| | **6** | **7** | **8** | **9** |
| Accuracy | 0.88 | 0.90 | 0.90 | 0.93 |
| Precision | 0.89 | 0.91 | 0.92 | 0.94 |
| Recall | 0.88 | 0.90 | 0.90 | 0.93 |
| F-Measure | 0.88 | 0.90 | 0.91 | 0.93 |

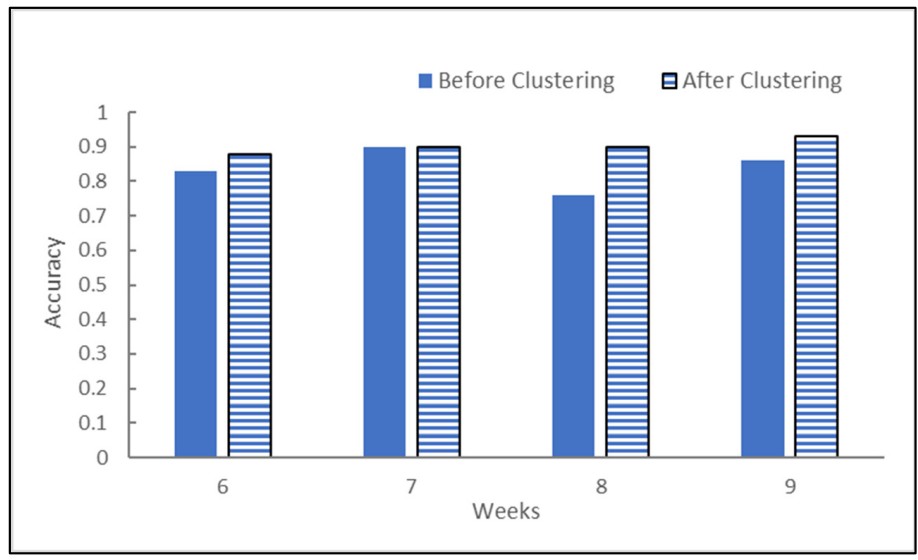

**Figure 6.** Accuracy before and after clustering.

## 5. Discussion

This study planned to improve on Sabri et al.'s [23] research by adding more features, such as demographic and learning program outcomes, that would help in identifying student patterns. By using K-means clustering, the student behavioral pattern was identified. There are mainly three types of students, which are low-, average-, and high-performance based on the learning program outcome feature. These students show different behavior performed by those groups throughout the weeks. Early in the semester, it shows that the student's behavior does not have a great impact on the student's performance by the end of the semester. However, in the last week, those with high self-efficacy and learning styles had high performance. Meanwhile, the student with low performance had average to low self-efficacy and learning style. A student's behavior on week 9 is the most important as it affects their performance the most.

In addition to that, research conducted by Sabri et al. [23] did not focus on the accuracy of the prediction model; therefore, the significance of the features used cannot be evaluated properly, unlike in research by Francis et al. [4]. This study has demonstrated that building

a new model based on clustering enhances the ability to predict student learning outcomes. The accuracy of the prediction model was higher than before the clustering, as shown in Figure 6. Week 8 had the biggest improvement in terms of accuracy, with 0.76 increasing to 0.90 after clustering. Therefore, using this combination of unsupervised and supervised learning is proven to give better results in terms of the accuracy of the prediction model.

## 6. Conclusions

Higher education institutions need to identify students' patterns and detect early at-risk students. The university and lecturers can then create preventive measures by identifying what characteristics at-risk students display and when is the most crucial time to identify students at risk. Students can start their semester with a great attitude but end with low performance or vice versa. Therefore, having a better understanding of student behavior is helpful. This paper was able to identify student patterns throughout the weeks in a semester by using K-means clustering. The dataset used has 140 students overall after cleaning, with 41 attributes consisting of demographic, self-efficacy, learning style, and learning program outcome. By using K-means clustering, the learning program outcome became the main feature contributing to the clusters created. Three clusters were created according to the low, average, and high learning program outcome mean values. The three clusters were compared from week 6 to week 9. Week 9 impacted the most on learning program outcomes; those with high self-efficacy and learning style had a high value for the learning program outcome. Before clustering, the dataset was labeled according to the student's semester grades. After categorizing the student dataset according to the cluster using K-means clustering, a new prediction model was developed. A comparison has been made between the prediction model, using the semester grade as the label and the learning program outcome as a label. The prediction model using the clusters label obtained a higher accuracy. Therefore, K-means clustering is beneficial in predicting a student's performance and is useful in identifying a student's pattern throughout a semester. K-means clustering will be improved in future research to further enhance the outcomes by refining the pre-processing procedure and experimenting with different cluster numbers. In future works, related to the insignificant result between clusters, the size of the dataset could be further increased to give better accuracy of the prediction model. In addition to that, adding a feature selection method can help reduce the number of features and identify the important features that contribute to a student's performance.

**Author Contributions:** Conceptualization, N.A.A.M.; formal analysis, N.I.M.T.; funding acquisition, N.A.A.M.; investigation, N.I.M.T.; methodology, N.I.M.T.; project administration, N.A.A.M.; resources, N.A.A.M.; software, N.I.M.T.; supervision, N.A.A.M. and S.S.; validation, N.I.M.T.; writing—original draft, N.I.M.T.; writing—review and editing, N.A.A.M. and S.S. All authors have read and agreed to the published version of the manuscript.

**Funding:** This research was funded by the Universiti Kebangsaan Malaysia: GUP-2020-090 and DIP-2019-013.

**Institutional Review Board Statement:** Not applicable.

**Informed Consent Statement:** Not applicable.

**Data Availability Statement:** Not applicable.

**Conflicts of Interest:** The authors declare no conflict of interest.

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
