# Peer review of "Identification of Student Behavioral Patterns in Higher Education Using K-Means Clustering and Support Vector Machine"

_applsci, doi:10.3390/app13053267_

Round 1

Reviewer 1 Report

The manuscript developed a prediction model for assessing student performance by combining classification and clustering methods. The model with the new labels gained higher accuracy when compared to the student dataset with labels using their semester grade. The study contributes valuable insights that higher education institutions may utilize. However, some problems need to be corrected before the manuscript can be accepted.

(1) Give the full name of the abbreviation (e.g., IR4.0 in line 37, CGPA in line 121, and Support Vector Machines in line 244) when it first appears and uses the abbreviation in the rest of this article. The author should ensure that all abbreviations have been accurately checked.

(2) For consistency, the author should unify the citation format across the manuscript following the journal requirements (e.g., "study of Aggarwal et al. (2021)" in line 103 and "the study of [17]" in line 124).

(3) Should be "were handed" in line 191.

(4) "One advantage of employing SVM is that it works well in high-dimensional environments, even when there are more dimensions than samples." The authors should cite relevant literature to prove this statement.

(5) Should be "Figure 4" in line 272.

(6) The data in Tables 7 and 8 can be presented in bar charts, which are more visual.

(7) Should be "students show" in line 305.

Reviewer 2 Report

  This manuscript proposed a machine learning model to predict academic performance of students. The authors combined k-means clustering for effective labeling and SVM for classification. The results are interesting but there are some comments for revision.

3.1 Data preparation,

- The authors extracted 41 features using a survey. How could you quantify each variable (Table 3- Table 6)?

3.2 Data pre-processing,

- They removed some datasets that were lots of empty information. Is there any criterion for removing datasets? After elimination, what is the final number of datasets?

3.3 Clustering,

- A Figure related to elbow rule and determination of the cluster number 3 should be included for better explanation.

- What are advancements of results, compared to reference 22?

- Figure 4: both mean and standard deviation values should be added.

- Table 6: HPP 6, 7 -> HPP 5, 6

- Please check indentation, carefully.

- line 121: What is CGPA?

- line 272, 275: Figure 5 --> Figure 4

Reviewer 3 Report

The article describes a study on the ability to predict student performance in order to improve academic performance.

The aim of the study is to identify features that form clusters with holistic characteristics and to develop and validate a prediction model for each of the clusters to predict student performance holistically.

The study is well written and easy to read, with the necessary description for the study under review.

Round 2

Reviewer 2 Report

Thank you for kind responses. However, there are some points to be revised.

- Figure 2 was newly added in the revised manuscript. However, k=3 don't seem like elbow. The slope in the figure did not sharply change where k=3.

- Figure 5 was changed in the revised manuscript. The values of standard deviations were very large so it seems there are no statistical differences. Could you try t-test among values?
